# Impact of COVID-19-related knowledge on protective behaviors: The moderating role of primary sources of information

Sooyoung Kim[1], Ariadna Capasso[2], Stephanie H. Cook[2,3], Shahmir H. Ali[2], Abbey M. Jones[4], Joshua Foreman[2], Ralph J. DiClemente[2], Yesim Tozan[1,5]*

1 Department of Public Health Policy and Management, New York University School of Global Public Health, New York, New York, United States of America, 2 Department of Social and Behavioral Sciences, New York University School of Global Public Health, New York, New York, United States of America, 3 Department of Biostatistics, New York University School of Global Public Health, New York, New York, United States of America, 4 Department of Epidemiology, New York University School of Global Public Health, New York, New York, United States of America, 5 Global and Environmental Public Health Program, New York University School of Global Public Health, New York, New York, United States of America

* tozan@nyu.edu

**Data Availability Statement:** All relevant data are within the manuscript and its Supporting Information files.

## Abstract

This study assessed the modifying role of primary source of COVID-19 information in the association between knowledge and protective behaviors related to COVID-19 among adults living in the United States (US). Data was collected from 6,518 US adults through an online cross-sectional self-administered survey via social media platforms in April 2020. Linear regression was performed on COVID-19 knowledge and behavior scores, adjusted for sociodemographic factors. An interaction term between knowledge score and primary information source was included to observe effect modification by primary information source. Higher levels of knowledge were associated with increased self-reported engagement with protective behaviors against COVID-19. The primary information source significantly moderated the association between knowledge and behavior, and analyses of simple slopes revealed significant differences by primary information source. This study shows the important role of COVID-19 information sources in affecting people's engagement in recommended protective behaviors. Governments and health agencies should monitor the use of various information sources to effectively engage the public and translate knowledge into behavior change during an evolving public health crisis like COVID-19.

## Introduction

The World Health Organization (WHO) declared COVID-19 a Public Health Emergency of International Concern (PHEIC) on January 30, 2020 [1]. The COVID-19 pandemic continues to cause significant global disruption across sectors ranging from healthcare to education to the economy. Millions of people around the world have been subjected to mitigation strategies, including stay-at-home restrictions, physical and social distancing, and mask wearing, all of which have necessitated substantial changes in individual behaviors for the collective good of

**Funding:** The authors received no specific funding for this work.

**Competing interests:** The authors have declared that no competing interests exist.

the community. Despite recommendations by public health authorities, people have had varying levels of engagement with protective behaviors against COVID-19, and this has presented a major obstacle to the success of measures to control the pandemic [2–5].

Knowledge is shown to be positively associated with health protective behaviors [6–8]. During the current pandemic, an associated infodemic, defined as "an overabundance of information—some accurate and some not—"has made it challenging for people to find reliable and credible sources to acquire knowledge when they need it [9]. The start of COVID-19 as pneumonia of unknown aetiology [10] allowed for extensive speculation of the origins of the disease and the limited information at the time [11]. This led to delays in communication of the scientific knowledge about the disease by public health authorities. In addition, the rise of online communication mediums, such as social media, blogs, and podcasts, has resulted in user-generated content that is often disseminated without verification of its veracity and consumed at an unprecedented speed and scale by the public. A study conducted in six developed countries in April 2020 showed that while the majority of people used official news organizations as their primary source of information, about half of the participants reported also using Google or other online search and social media platforms for COVID-19-related information. Specifically, 25–53% of the participants across six countries reported using Facebook to obtain information on COVID-19 at least once over the past week, while 15–46% of the participants used YouTube for the same purpose [12].

The current challenge for public health authorities is, therefore, to strategize the dissemination of COVID-19-related information to counter the misinformation emanating from these easily accessible and often unregulated online sources and to deliver timely and correct information to the public supported by scientific evidence. During this pandemic, evidence has shown that people use a vast range of sources to get COVID-19-related information [13], and their choice of primary source reflects their trust in the legitimacy of these sources and affects their attitudes and vaccine uptake, as also supported by past research on vaccine hesitancy in general [14, 15]. Despite the growing volume of research on the COVID-19 infodemic, evidence is lacking on how the source from which people acquire COVID-19-related knowledge influences the association between knowledge and protective behaviors. Prior to COVID-19, the use of social media, including Wikipedia, blogs, and social networking services platforms, as a source of health-related information to affect behavior change was found effective but at the same time risky because of the trustworthiness of information depending on the information-seeking context and source [16–18]. Furthermore, no study to date has examined the role of different information sources in moderating the association between knowledge and behaviors while controlling for other factors.

In this study, we tested the hypothesis that *different primary sources of COVID-19 information will act as an effect modifier of the relationship between levels of knowledge and self-reported engagement in protective behaviors*. We focused on the primary source of information based on the idea that people's choice of primary information source is a manifestation of their health consciousness and motivation for health-oriented behaviors [19]. The findings of this study have the potential to improve the targeting and effectiveness of risk communication strategies seeking to achieve behavior change.

## Methods

### Study participants and design

This study used data from an online survey conducted in April 2020. Details of the survey design and administration are reported elsewhere [20]. Briefly, the sample was recruited among social media users through an online advertisement campaign within Facebook and its

affiliated platforms. The eligibility was limited to English-speaking adults aged 18 years and over residing in the United States (US). Participation was voluntary, and participants did not receive any compensation. For this study, we only included participants who provided written informed consent and responded to all behavior- and knowledge-related questions. As a result, a total of 6,518 responses were included in the analysis. The study protocol was reviewed and deemed exempt by the affiliated institution's Institutional Review Board.

## Questionnaires and variables of interest

The questionnaire was designed based on the Health Belief Model (HBM) [21] and the World Health Organization (WHO) survey tool for behavioral insights on COVID-19 [22].

**Outcome variable.** The outcome of interest was the degree of self-reported engagement with recommended protective behaviors against COVID-19. For this, an index variable was derived from participants' answers to a set of 13 binary questions. Answers were assessed based on the recommendations on protective behaviors provided by the Centers for Disease Control and Prevention (CDC). While correct answers were assigned a score of 1, answers that do not comply with the CDC recommendations at the time of the survey were scored 0. The sum of scores was used as the overall behavior score (Range: 0–13). Cronbach's alpha coefficient was used to assess the internal consistency of the summed scores and was 0.7 and was deemed acceptable [23]. All the questions used to construct the outcome variable are provided in the S1 Table.

**Predictor variables.** The main predictor was the level of knowledge on COVID-19. Similar to the behavior score, we created an index variable for the knowledge score using participants' answers to a set of 21 binary questions. An overall knowledge score (Range: 0–21) was calculated by assigning 1 for the correct answers and 0 for the incorrect answers. We tested the internal consistency of the questions used to derive the sum of scores and Cronbach's alpha coefficient was 0.60 for the knowledge score and was deemed acceptable [23]. All the questions used to construct the predictor variable are provided in the S1 Table.

**Moderator variables.** Participant's primary source of COVID-19-related information was used as a moderator. The questionnaire asked participants the primary source they used to acquire COVID-19-related information, with the ability to choose only one answer from a list of choices. Primary information sources were categorized into six mutually exclusive categories: (1) family, friends, and colleagues; (2) doctor or medical provider; (3) government or other official sources (e.g., CDC or WHO); (4) traditional media (e.g., newspapers, TV); (5) new media (e.g., social media, web surfing on non-official sources, podcasts); and (6) religious leaders.

## Statistical analysis

All statistical analyses were conducted in R (version 3.6.3). The minimal dataset to replicate the analysis is provided in the S1 Dataset. First, we stratified the participants into groups by their primary source of information. We then used chi-square test to evaluate differences in demographic characteristics by group. Pairwise Wilcoxon rank-sum test—a nonparametric alternative to the *t* test—was used, given the left-skewedness of the scores' distribution, to compare the distribution of knowledge and behavior scores between each group.

In order to test the hypothesis, ordinary least squares (OLS) linear regression was performed with behavior score as dependent variable, knowledge score as independent variable, and with an interaction term between knowledge score and primary source of information. Sociodemographic factors, including age, sex, employment status, education level, number of sources to acquire COVID-19-related information, and political affiliation, which may affect

self-reported engagement with recommended protective behaviors, represented by vector Z in the equation below, were included as covariates [13, 24]. We reported adjusted regression coefficients and corresponding *p*-values and 95% confidence intervals (CIs).

$$
\begin{aligned}
(Protective\ behavior) \\
= \beta_0 + \beta_1 * Knowledge + \beta_2 * Knowledge * (Primary\ source\ of\ information) + \beta_3 \\
* (Primary\ source\ of\ information) + \beta_4 * Z + \epsilon
\end{aligned} \tag{1}
$$

## Results

### Description of sample

A total of 6,518 people participated in the survey from April 16 to 21, 2020. Of which, 1,984 (36.8%) participants indicated government or other official sources as their primary source of COVID-19-related information, followed by 1,792 (33.9%) who reported doctor or medical provider (Table 1). Traditional media channels were preferred by 721 (13.6%), and new media by 519 (9.8%) participants, whereas a small fraction (309; 5.8%) cited family, friends, or colleagues. Only three participants named religious leaders as their primary information source. Chi-square test showed almost all demographic variables, including age, sex, race, employment status, and political affiliation (p-value <0.01) were significantly differently distributed between each group, as shown in Table 1. Participants in older age groups preferred doctors or traditional media, whereas those who self-identified as non-white (*p*-value <0.001), Republican (*p*-value <0.001), or residing in a rural area (*p*-value = 0.011) were over-represented among those who indicated social media or family, friends, and colleagues as their primary information source. Standardized residuals of the chi-square test are reported in the S2 Table.

The highest knowledge score was observed for participants who used traditional media (Median = 20, IQR 19–21), government or official sources (Median = 20, IQR 19–21), and doctors or medical providers as primary information source (Median = 20, IQR 19–20), followed by those who preferred new media (Median = 19, IQR 18–20) and family, friends, and colleagues (Median = 19, IQR 17–20). A similar trend was observed for the behavior score with less variability across primary information sources.

### Regression analysis on the association between level of COVID-19 knowledge and degree of protective behaviors

The main effect model without the interaction term (adjusted R-square = 0.212), and the fully adjusted regression model with the moderator are reported in Table 2 (adjusted R-square = 0.224). In the main effect model, higher level of knowledge score was positively associated with higher degrees of engagement with protective behaviors against COVID-19. When controlled for all covariates, a unit increase in the knowledge score was associated with a 0.273 increase in the behavior score (95% CI: 0.241–0.305, p-value<0.01).

In the fully adjusted model, all interaction terms between the knowledge score and the primary source of information were significantly associated with the changes in the behavioral score (excluding religious leaders because a coefficient could not be derived due to the limited sample size (n = 3)). In summary, while the behavior score was positively associated with the knowledge score (adjusted coefficient 0.275, p-value <0.01), when all covariates were held constant, the association was significantly stronger when the primary source of information was social media, podcasts or unofficial websites (interaction term coefficient 0.1, p-value = 0.031), or family, friends and colleagues (interaction term coefficient 0.158, p-value <0.01), in comparison to when the primary source was through doctor or medical staff (reference category). On the contrary, the association was significantly weaker when the primary

**Table 1. Demographics of the study participants (n = 6,518).**

| | Total (n = 6518) | Doctor or medical provider (n = 1792) | Government or other official sources (e.g. CDC or WHO) (n = 1948) | Traditional media (n = 721) | New media (Social media, web surfing, podcasts, and etc. (n = 519) | Family, friends, and coworkers (n = 309) | Religious leaders (n = 3) | p-value |
|---|---|---|---|---|---|---|---|---|
| **Sex** | | | | | | | | <0.001 |
| Female | 3717 (57.6%) | 975 (54.8%) | 1289 (66.8%) | 448 (62.8%) | 225 (43.7%) | 139 (45.4%) | 3 (100.0%) | |
| Male | 2738 (42.4%) | 804 (45.2%) | 641 (33.2%) | 265 (37.2%) | 290 (56.3%) | 167 (54.6%) | 0 (0.0%) | |
| **Age group** | | | | | | | | <0.001 |
| 18–39 years old | 1078 (16.5%) | 243 (13.6%) | 389 (20.0%) | 107 (14.8%) | 82 (15.8%) | 51 (16.5%) | 0 (0.0%) | |
| 40–59 years old | 2811 (43.1%) | 770 (43.0%) | 876 (45.0%) | 264 (36.6%) | 262 (50.5%) | 140 (45.3%) | 2 (66.7%) | |
| 60+ years old | 2629 (40.3%) | 779 (43.5%) | 683 (35.1%) | 350 (48.5%) | 175 (33.7%) | 118 (38.2%) | 1 (33.3%) | |
| **Race** | | | | | | | | 0.006 |
| White, Non-Hispanic | 6012 (92.2%) | 1675 (93.5%) | 1817 (93.3%) | 679 (94.2%) | 469 (90.4%) | 277 (89.6%) | 2 (66.7%) | |
| Non-White | 506 (7.8%) | 117 (6.5%) | 131 (6.7%) | 42 (5.8%) | 50 (9.6%) | 32 (10.4%) | 1 (33.3%) | |
| **Employment status** | | | | | | | | <0.001 |
| Employed | 2845 (56.2%) | 941 (54.9%) | 1096 (58.4%) | 354 (51.1%) | 300 (61.6%) | 153 (53.5%) | 1 (50.0%) | |
| Student/Unpaid work | 280 (5.5%) | 74 (4.3%) | 128 (6.8%) | 37 (5.3%) | 21 (4.3%) | 20 (7.0%) | 0 (0.0%) | |
| Not working/ Unemployed | 635 (12.5%) | 204 (11.9%) | 239 (12.7%) | 87 (12.6%) | 66 (13.6%) | 39 (13.6%) | 0 (0.0%) | |
| Retired | 1300 (25.7%) | 495 (28.9%) | 415 (22.1%) | 215 (31.0%) | 100 (20.5%) | 74 (25.9%) | 1 (50.0%) | |
| **Educational attainment** | | | | | | | | 0.0602 |
| High school or less | 516 (13.9%) | 178 (13.8%) | 190 (14.0%) | 49 (10.8%) | 62 (15.9%) | 37 (17.3%) | 0 (0.0%) | |
| Some college / Associate's degree | 1720 (46.5%) | 626 (48.6%) | 613 (45.3%) | 198 (43.7%) | 177 (45.5%) | 105 (49.1%) | 1 (100.0%) | |
| Bachelor's degree or higher | 1463 (39.6%) | 484 (37.6%) | 551 (40.7%) | 206 (45.5%) | 150 (38.6%) | 72 (33.6%) | 0 (0.0%) | |
| **Political affiliation** | | | | | | | | <0.001 |
| Democrat | 1925 (38.3%) | 610 (35.7%) | 756 (40.4%) | 397 (57.9%) | 96 (20.1%) | 66 (23.2%) | 0 (0.0%) | |
| Republican | 1222 (24.3%) | 417 (24.4%) | 425 (22.7%) | 109 (15.9%) | 161 (33.8%) | 108 (37.9%) | 2 (100.0%) | |
| Other | 1072 (21.3%) | 382 (22.4%) | 403 (21.6%) | 98 (14.3%) | 132 (27.7%) | 57 (20.0%) | 0 (0.0%) | |
| Prefer not to say | 809 (16.1%) | 299 (17.5%) | 286 (15.3%) | 82 (12.0%) | 88 (18.4%) | 54 (18.9%) | 0 (0.0%) | |

source of information was traditional media (interaction term coefficient -0.109, p-value = 0.044), or the government or other official sources (interaction term coefficient -0.096, p-value = 0.018). For all primary information sources, the increase in the behavioral score was larger with the increasing number of sources used (adjusted coefficient 0.156, 95% CI: 0.132–0.179, p-value<0.01). The F-test showed that model with interaction term fitted the data significantly better than the main effect model (p-value <0.01).

**Table 2. Main effect model and the full linear regression model between the COVID-19 knowledge score and the protective behavior score with covariates and the interaction term (n = 3,663).**

| Variables | Main effect model | | | Model with interaction term | | |
|---|---|---|---|---|---|---|
| | Coefficient | 95% Confidence interval | p-value | Coefficient | 95% Confidence interval | p-value |
| **(Intercept)** | 2.986 | (2.336, 3.637) | <0.001 | 2.967 | (1.915, 4.020) | <0.001 |
| **Knowledge score** | 0.273 | (0.241, 0.305) | <0.001 | 0.275 | (0.221, 0.328) | <0.001 |
| **Source of information** | | | | | | |
| Doctor or medical staff | Ref | | | Ref | - | - |
| Government or other official sources (e.g., CDC or WHO) | 0.146 | (0.023, 0.269) | 0.02 | 2.021 | (0.481, 3.561) | 0.01 |
| Traditional media | 0.182 | (0.010, 0.354) | 0.038 | 2.297 | (0.233, 4.361) | 0.029 |
| New media (Social media, web surfing, podcasts, etc. | -0.302 | (-0.484, -0.120) | 0.001 | -2.155 | (-3.863, -0.447) | 0.013 |
| Family, friends, and coworkers | -0.299 | (-0.532, -0.066) | 0.012 | -3.174 | (-5.094, -1.255) | <0.001 |
| Religious leader | -2.514 | (-5.613, 0.586) | 0.112 | -2.508 | (-5.595, 0.580) | 0.111 |
| **Political affiliation** | | | | | | |
| Democrat | Ref | | | Ref | - | - |
| Republican | -0.39 | (-0.528, -0.251) | <0.001 | -0.393 | (-0.531, -0.255) | <0.001 |
| Other | -0.329 | (-0.473, -0.184) | <0.001 | -0.335 | (-0.478, -0.191) | <0.001 |
| Prefer not to say | -0.163 | (-0.318, -0.008) | 0.04 | -0.181 | (-0.336, -0.026) | 0.022 |
| **Number of sources** | 0.158 | (0.134, 0.181) | <0.001 | 0.156 | (0.132, 0.179) | <0.001 |
| **Age group** | | | | | | |
| 18–39 years old | Ref | | | Ref | | |
| 40–59 years old | 0.148 | (-0.005, 0.301) | 0.058 | 0.152 | (0.000, 0.305) | 0.05 |
| 60+ years old | 0.182 | (0.002, 0.363) | 0.047 | 0.189 | (0.009, 0.368) | 0.039 |
| **Sex** | | | | | | |
| Female | Ref | | | Ref | - | - |
| Male | -0.502 | (-0.610, -0.395) | <0.001 | -0.505 | (-0.612, -0.398) | <0.001 |
| **Educational attainment** | | | | | | |
| High school degree or lower | Ref | | | Ref | - | - |
| Some college / Associate degree | -0.312 | (-0.475, -0.148) | 0.005 | -0.214 | (-0.371, -0.057) | 0.007 |
| Bachelor's degree or higher | -0.223 | (-0.380, -0.066) | <0.001 | -0.301 | (-0.464, -0.138) | <0.001 |
| **Employment status** | | | | | | |
| Employed | Ref | | | Ref | - | - |
| Student/Unpaid work | 0.078 | (-0.148, 0.304) | 0.499 | 0.063 | (-0.163, 0.288) | 0.586 |
| Not working/Unemployed | 0.417 | (0.266, 0.569) | <0.001 | 0.416 | (0.265, 0.567) | <0.001 |
| Retired | 0.208 | (0.056, 0.360) | 0.007 | 0.199 | (0.047, 0.351) | 0.01 |
| **Interaction term** | | | | | | |
| Knowledge score * D&M* | | | | Ref | - | - |
| Knowledge score * GOV* | | | | -0.096 | (-0.175, -0.017) | 0.018 |
| Knowledge score * TRAD* | | | | -0.109 | (-0.215, -0.003) | 0.044 |
| Knowledge score * NEWM* | | | | 0.1 | (0.009, 0.190) | 0.031 |
| Knowledge score * FFC* | | | | 0.158 | (0.055, 0.262) | 0.003 |

*D&M: Doctors or medical staff/ GOV: Government or other official sources / TRAD: Traditional media / NEWM: New media (Social media, web surfing, podcasts, etc.) / FFC: Family, friends and coworkers.

The visualization of this result as a fitted linear plot of the association between COVID-19 knowledge and behavior with the modifying effect of primary information source is shown in Fig 1. In this figure, the slope represents the association between knowledge and behavior. The varying slopes by different primary information source, as represented by different colors in

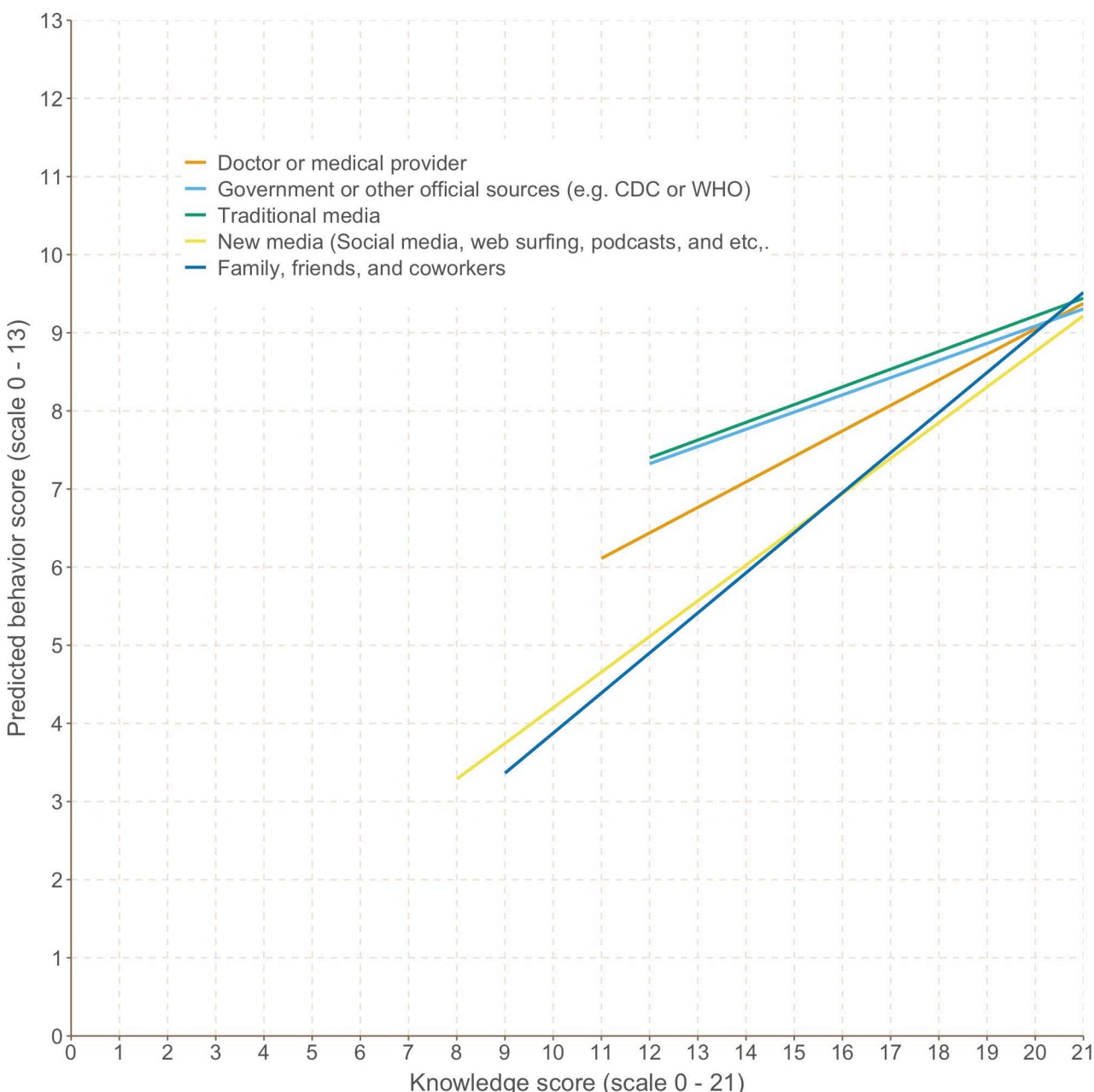

**Fig 1. Fitted linear model for the association between COVID-19 knowledge (x-axis) and protective behaviors (y-axis) by primary information source of COVID-19.**

the legend, corresponds to the effect modification as shown by the significant coefficients for interaction terms in Table 2.

## Discussion

Our study provides one of the first empirical evidence on effect modification by primary source of information on the association between knowledge and engagement with protective behaviors against COVID-19. The findings can be summarized in three points: First, the primary source of COVID-19 knowledge differs across sociodemographic subgroups, which may

result in varying levels of knowledge related to COVID-19. Distinctively, participants with the lowest level of knowledge preferred informal sources, such as social media and family, friends and colleagues, as their primary source of information. Second, among those with high levels of knowledge, the primary source of information did not predict their protective behaviors. Lastly, the primary source of information significantly moderated the association between knowledge and behavior, and analyses of simple slopes revealed significant differences by primary information source (Fig 1). While our study design is cross-sectional and therefore does not allow us to draw any inference on causality, the results suggest that different communication media may deliver the same information in distinctive ways, which may lead to differing levels of knowledge across individuals and translate into different levels of engagement with behaviors.

This study brings particular attention to two sources of information: 1) online media sources, including social media and websites other than those of official governments or international organizations, and 2) informal communication between family, friends, and colleagues. In contrast to other information sources, acquisition of knowledge from both of these sources occurs interactively and informally. Not only were both a preferred source of COVID-19-related information among the study participants at the bottom quartile of the knowledge score distribution, but also these sources were associated with a higher increase in the level of engagement with protective behaviors given the same unit increase in the knowledge score, as demonstrated by steeper slopes in Fig 1. These two contrasting findings from our study suggest that these sources, when leveraged well, hold potential to empower people. By the same token, when misused, these information sources may present a health threat. The mechanism of how information presented through these informal sources can potentially be associated with behavior change could be an area of future research. Due to the limitation of the cross-sectional study design, it is not within the scope of the current study to test whether these two sources had indeed diminished the level of both correct knowledge and engagement with protective behaviors. Regardless, the results suggest that, when these sources are leveraged appropriately to improve knowledge, the translation of knowledge to behavior among the participants who primarily uses these sources could be effective.

In summary, primary sources of information may be partially accountable for varying levels of COVID-19-related knowledge, reflecting different sociodemographic characteristics of the main audience of each source, and its heterogeneous associations with individuals' engagement with protective behaviors against COVID-19. Our results suggest that the primary source of information may act as a moderator in the pathway from knowledge to behavior, and sources of information and the manner in which each source conveys information to the public could serve as the tangible target of intervention for improved risk communication.

The study has a number of limitations that leave room for further research. First, the study design introduces a number of biases. Despite the large sample size, the study sample, drawn from nonprobability convenience sampling via social media platforms affiliated with Facebook, is not representative of the US population [20]. While our sample showed a balanced distribution of participants from every US state, age group, and type of residence, certain key subpopulations are underrepresented in the study sample. For example, the survey did not include the people without access to Internet or social media account affiliated with Facebook. While around 70% of the US population are estimated to have Facebook accounts, and among them, 75% use Facebook on a daily basis [25], it is also estimated that about 20% of the US households do not have Internet at home [26]. In addition, more than 40 million adults in the US are of foreign origin, and almost one third of them do not speak English well, thus would be unable to participate in our study [27]. Our sample of participants was also overwhelmingly non-Hispanic white. Overall, high scores of COVID-19-related knowledge observed in this

study could have been due to the sampling strategy, which most likely attracted people who were more interested in COVID-19. Meanwhile, the small sample size of those who reported seeking COVID-19-related information primarily from religious leaders could be due to the lack of engagement of this specific group in social media platforms in general [28]. To partially overcome this limitation, we made substantial efforts to oversample from potentially under-represented groups, including men and racial and ethnic minorities [20]. However, it is important to note that, due to all of the aforementioned factors, the findings from our study are not generalizable to the US population as a whole. We still strongly believe that our findings hold important implications for designing risk communication strategies, targeting particularly those who use social media platforms as their primary source of information.

Second, level of engagement with protective behaviors was based on self-report, which is subject to response bias. Although presented in random order, the questions on engagement with protective behaviors were relatively simple and dichotomized, and it was straightforward to guess the correct answers; this may have resulted in over-estimation of the level of engagement in specific subgroups. Moreover, given the nature of the observational design, our study is subject to many known and unknown confounders, which should be addressed by quasi-experimental or experimental studies.

Additionally, measurements of the level of knowledge and behavioral engagement were done via questions developed and used in previous surveys [20], but we used unvalidated scales based on the sum of correct answers to the survey questions without being able to test their validity. Internal consistency of the questions was tested and shown to be reliable for the behavior score, but less reliable for the knowledge score. Future studies using validated scales to measure behaviors and knowledge are needed.

Lastly, overall high knowledge scores among the study participants and the resulting lack of variability in the independent variable may be regarded as a limitation to the analysis. A supplementary qualitative study of the participants could facilitate a deeper understanding of the association between knowledge and behaviors in order to better inform risk communication strategies.

## Conclusions

As the COVID-19 pandemic persists over a year since its first detection, more and more people are feeling fatigued about continuous engagement with protective measures [29, 30]. However, it is crucial to maintain public vigilance and a collectively high level of engagement with protective behaviors in order to prevent further spread of the virus, particularly because of the slow progress with COVID-19 vaccine implementation and the emergence of several variants of the COVID-19 virus [31]. Even small deficiencies in protective behavior compliance can result in significant community spread, as evidenced by the second and third waves of COVID-19 cases in several countries, including the US [32, 33]. With the concurrent infodemic severely hindering the ability of health authorities to convey timely and accurate information, customizing risk communications with appropriate prioritization of high-risk populations and information platforms with stronger capacity to promote behavior change (such as online or interpersonal information sources) is of absolute importance. In order to effectively promote community-level protective behaviors, it is necessary to step out of the traditional way of mass-scale and didactic communication and proactively reach out and engage through the channels where people are more willing to seek information.

## Supporting information

**S1 Table. Questionnaire used in the survey.**
(DOCX)

**S2 Table. Standardized residuals of chi-square test performed on the demographics of the study participants (n = 6,518).**
(DOCX)

**S1 Dataset. Study dataset.**
(CSV)

## Author Contributions

**Conceptualization:** Sooyoung Kim, Yesim Tozan.

**Data curation:** Ariadna Capasso, Shahmir H. Ali, Abbey M. Jones, Joshua Foreman, Ralph J. DiClemente, Yesim Tozan.

**Formal analysis:** Sooyoung Kim.

**Investigation:** Sooyoung Kim, Ariadna Capasso.

**Methodology:** Sooyoung Kim, Stephanie H. Cook.

**Supervision:** Ralph J. DiClemente, Yesim Tozan.

**Validation:** Sooyoung Kim, Ariadna Capasso, Stephanie H. Cook, Shahmir H. Ali, Abbey M. Jones, Joshua Foreman, Yesim Tozan.

**Visualization:** Sooyoung Kim.

**Writing – original draft:** Sooyoung Kim.

**Writing – review & editing:** Sooyoung Kim, Ariadna Capasso, Stephanie H. Cook, Shahmir H. Ali, Abbey M. Jones, Joshua Foreman, Ralph J. DiClemente, Yesim Tozan.

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
