## [Decision Letter · Decision Letter 0]

8 Sep 2021

PONE-D-21-24232Impact of COVID-19-Related Knowledge on Protective Behaviors: The Moderating Role of Primary Sources of InformationPLOS ONE

Dear Dr. Tozan,

Thank you for submitting your manuscript to PLOS ONE. After careful consideration, we feel that it has merit but does not fully meet PLOS ONE’s publication criteria as it currently stands. Therefore, we invite you to submit a revised version of the manuscript that addresses the points raised during the review process.

In the revised version of the paper, please emphasize more on the methodological aspects. Please provide a figure in which the methodology is presented and explain it in more words within the body of the paper. Also, please consider to add more information in the results section, commenting upon the role of multiple sources of information.

We look forward to receiving your revised manuscript.

Kind regards,

Camelia Delcea

Academic Editor

PLOS ONE

Journal Requirements:

2. Please ensure that you have specified (1) whether consent was informed, (2) what type you obtained (for instance, written or verbal, and if verbal, how it was documented and witnessed). If your study included minors, state whether you obtained consent from parents or guardians. If the need for consent was waived by the ethics committee and (3) If you are reporting a retrospective study of medical records or archived samples, please ensure that you have discussed whether all data were fully anonymized before you accessed them and/or whether the IRB or ethics committee waived the requirement for informed consent. If patients provided informed written consent to have data from their medical records used in research, please include this information.

Reviewers' comments:

Reviewer's Responses to Questions

**Comments to the Author**

1. Is the manuscript technically sound, and do the data support the conclusions?

Reviewer #1: Partly

Reviewer #2: Yes

Reviewer #3: No

Reviewer #4: No

2. Has the statistical analysis been performed appropriately and rigorously? 

Reviewer #1: Yes

Reviewer #2: No

Reviewer #3: Yes

Reviewer #4: No

3. Have the authors made all data underlying the findings in their manuscript fully available?

Reviewer #1: No

Reviewer #2: No

Reviewer #3: Yes

Reviewer #4: Yes

4. Is the manuscript presented in an intelligible fashion and written in standard English?

Reviewer #1: Yes

Reviewer #2: Yes

Reviewer #3: Yes

Reviewer #4: Yes

5. Review Comments to the Author

Reviewer #1: Review for PONE-D-21-24232

The study explores the moderating effects of information sources on the association between COVID-19 knowledge and behavior. The study is interesting and relevant for COVID-19-related interventions but also the field of health communication more generally. I have several suggestions that I believe would help make the manuscript clearer to the reader. Otherwise, I believe this is a good paper.

1. The introduction seems to focus on the effects of information sources on health behavior in the context of COVID-19. I think the study would benefit from a review of this topic in other contexts; if none exist, this could also be emphasized as a further contribution of the current study.

2. On page 2 line 55, “6-53% of participants…” this seems like a wide range, and it is unclear what the numbers refer to. Are they different numbers for different platforms?

3. Usually, sample size and reliabilities are mentioned in the Methods section rather than the results. The reliabilities also seem to be rather weak, perhaps removing some items could improve them?

4. On page 4 line 82, what is HBM?

5. On page 4, line 89, by “incorrect” do you mean not meeting CDC guidelines? This is not “incorrect” in the usual sense of the word. I also think the authors should add the scales as an appendix or at least give an example of the items because right now it is difficult to understand what behaviors are included.

6. I suggest the authors use a Rasch model to score the items rather than simple dichotomous scores because different items should contribute differently to the final score. A person who refrains from leaving their house at all but does not wash their hands often is different from a person who washes their hands but does not wear a mask when meeting others.

7. Why did the authors decide to have the information sources mutually exclusive? Since the authors have data on the number of sources, it would have made more sense to allow participants to choose several sources.

8. Why was it not possible to use a t-test?

9. The authors provide good details on their analysis.

10. I suggest adding standardized residuals to the chi-square analyses so it is clearer what categories were more extremely represented in each group.

11. On page 13, lines 193-195, “the primary source of information did not significantly moderate the association between knowledge and behaviors among the population with high levels of knowledge on COVID-19” – I understand what the authors wanted to say but the phrasing here is wrong. Of course there was no moderation among people with high levels of knowledge, as there is no variance among them. Should be “among those with high levels of knowledge, the primary source of information did not predict behavior”. Also, since the authors did not look at the population but at a sample, they should not refer to a population.

12. The authors conclude that the information sources lead to differing levels of knowledge which lead to different behaviors, and suggest that misusing certain sources can be dangerous. It seems to me that it is more likely that people who wanted to seek information were active about it and talked to doctors or government sources, whereas others heard about COVID-19 from sources they engage with anyway, namely, social media and personal relationships. Those who were interested in COVID-19 due to fear or other reasons looked into the topic, gained a lot of knowledge, and were more likely to engage with the recommended behaviors regardless of their source. I think the authors should be more careful when stating their conclusions.

13. While underrepresented demographic groups are certainly a limitation, the issue of representativeness of social media users seems more relevant here. The sample included Facebook users who are likely interested in COVID-19, otherwise, they wouldn’t have engaged with the study (this explains the high knowledge scores). This is likely to affect the results (e.g., people who turn to religious figures are less likely to participate, and if they do they probably do not represent other people who consult with religious authorities).

Reviewer #2: This study shows the important role of COVID-19 information sources in affecting 30 people’s engagement in recommended protective behaviors. The idea is well described theorotically and analytically.

Reviewer #3: While the findings of the study are interesting and comprehensive, one major shortcoming of the statistical analysis is that the authors have used inferential methods on the basis of a non-probability sample (convenience sample). As inferences are often drawn / valid only if a probability sample is used, so the authors need to justify it. If authors have followed a previous work where inferences have been drawn on the basis of a non-probability sample, they must cite it. The authors must clear this point.

Moreover, the abstract has been divided into sub-sections. I suggest the authors to write a brief abstract as a single section. No need to divide the abstract into sub-sections.

Further, in the text, citations have been given using parenthesis. Use square brackets for citations.

Reviewer #4: The paper addresses a "hot" topic in the context of the new coronavirus pandemic. The introduction provides even some information related to the literature review, even though the authors should try to better present the current state in a more critical manner. The methodology is quite simple as the data used in the paper is taken by the authors from external sources. The authors provide information related to predictor variables and moderator variables, but no scheme of the proposed model is provided. In my opinion, with given information, it is hard for the work to be reproduced or adapted to similar situations. The results section contains little information strictly related to the subject. The authors provide a long table containing descriptive statistics related to the participants to the study, with no real connection to the results as the results are very brief with respect to this information. Figure 1 is hard to understand - can you please explain it more in the main body of the paper? The moderator variable used in the study is a variable from a list. In most of the cases, it might happen that a person can be connected to multiple sources of information, each of them bringing its own contribution to the result. I think that a more-in-depth analysis is needed.

6. PLOS authors have the option to publish the peer review history of their article (what does this mean?). If published, this will include your full peer review and any attached files.

Reviewer #1: No

Reviewer #2: **Yes: **Rahim Alhamzawi

Reviewer #3: No

Reviewer #4: No

---

## [Author Response · Author response to Decision Letter 0]

30 Sep 2021

Response to Reviewer #1

The study explores the moderating effects of information sources on the association between COVID-19 knowledge and behavior. The study is interesting and relevant for COVID-19-related interventions but also the field of health communication more generally. I have several suggestions that I believe would help make the manuscript clearer to the reader. Otherwise, I believe this is a good paper.

-> Thank you for your thoughtful feedback. We hope the revisions improved the quality of our manuscript and addressed all the points you raised. 

1. The introduction seems to focus on the effects of information sources on health behavior in the context of COVID-19. I think the study would benefit from a review of this topic in other contexts; if none exist, this could also be emphasized as a further contribution of the current study.

-> Thank you for this suggestion. We updated the introduction with additional paragraph (line 71 – 76 to briefly cover the context prior to COVID-19 pandemic. 

2. On page 2 line 55, “6-53% of participants…” this seems like a wide range, and it is unclear what the numbers refer to. Are they different numbers for different platforms.

-> Thank you for pointing this out. We corrected the sentence to provide examples of varying social media and online search platform usage in 6 mentioned countries (lines 61-64). We hope the revised sentence reads better.

3. Usually, sample size and reliabilities are mentioned in the Methods section rather than the results. The reliabilities also seem to be rather weak, perhaps removing some items could improve them?

-> Thank you for this suggestion. We clarified the sample size (lines 90-91) and the measurement of internal consistency (lines 104-105 and lines 112-113) in the Methods section. For the reliability of the questions used for the predictor variable, we referenced Hulin et al.’s widely referenced work (https://www.jstor.org/stable/1480474?refreqid=excelsior%3Afbdac5eed827fb5efe04e1e fa5eb9616&seq=1#metadata_info_tab_contents), which considers the calculated value of 0.6 as acceptable. 

4. On page 4 line 82, what is HBM?

-> Thank you for pointing this out. We spelled out the Health Belief Model and added an appropriate reference introducing the HBM (line 95).

5. On page 4, line 89, by “incorrect” do you mean not meeting CDC guidelines? This is not “incorrect” in the usual sense of the word. I also think the authors should add the scales as an appendix or at least give an example of the items because right now it is difficult to understand what behaviors are included.

-> Thank you for this suggestion. We updated the term “incorrect answers” to “answers that does not comply with the CDC recommendations” in order to clarify (lines 101-104). We also presented the questions used to assess knowledge and behaviors in the Supplementary Material file (Table S1).

6. I suggest the authors use a Rasch model to score the items rather than simple dichotomous scores because different items should contribute differently to the final score. A person who refrains from leaving their house at all but does not wash their hands often is different from a person who washes their hands but does not wear a mask when meeting others.

-> Thank you for this suggestion. We appreciate the point you raised about how factor loading may be necessary to capture the different degrees of protection each behavior can exercise for an individual. Following your suggestion, we first used the Rasch model to understand the sensitivity to one’s ability and assess the misfit between persons and items, using the infit and its acceptable range of 0.75 and 1.33 (https://bookdown.org/ dkatz/Rasch_Biome/Rasch.html# optional---visualizing-item-fit). After running the model, we confirmed that the fit is consistent with the fit for the model utilizing the simple sum (see plots in the attached rebuttal letter). 

7. Why did the authors decide to have the information sources mutually exclusive? Since the authors have data on the number of sources, it would have made more sense to allow participants to choose several sources.

-> Thank you for this question. Our main hypothesis was that the “primary” source of information plays a significant moderating role in the relationship between knowledge and behavior. Also, the questionnaire was designed to inquire participants’ “primary” source of information, rather than the exhaustive list of all information sources used. However, in order to control for the confounding role of multiple information sources used by participants, we added a continuous variable on “total number of sources used” as a covariate in the analysis. 

8. Why was it not possible to use a t-test?

-> Thank you for this question. We used Wilcoxon rank sum test instead of t-test because t-test is a parametric test which requires samples meet certain pre-requirements including normality, equal variances, and independence. For both knowledge and behavior scores, the distribution was left skewed thus not meeting the normality assumption. Wilcoxon rank sum test is a widely-accepted non-parametric alternative test to t-test, which can be used when the sample distribution is not normal (https://bmcmedresmethodol.biomedcentral.com/articles/10.1186/1471-2288-12-78). 

9. The authors provide good details on their analysis.

-> Thank you for this positive feedback. 

10. I suggest adding standardized residuals to the chi-square analyses so it is clearer what categories were more extremely represented in each group.

-> Thank you for this suggestion. We reported the standardized residuals in the Supporting Information (Table S2) and indicated this information in the manuscript (line 159). 

11. On page 13, lines 193-195, “the primary source of information did not significantly moderate the association between knowledge and behaviors among the population with high levels of knowledge on COVID-19” – I understand what the authors wanted to say but the phrasing here is wrong. Of course there was no moderation among people with high levels of knowledge, as there is no variance among them. Should be “among those with high levels of knowledge, the primary source of information did not predict behavior”. Also, since the authors did not look at the population but at a sample, they should not refer to a population.

-> Thank you for this important comment. The sentence is now revised based on your input (line 220). We also checked all the sentences in the manuscript that used the word “population” and change the word to either “respondents” or “people” depending on the context so that it does not imply any inference to the general population.

12. The authors conclude that the information sources lead to differing levels of knowledge which lead to different behaviors, and suggest that misusing certain sources can be dangerous. It seems to me that it is more likely that people who wanted to seek information were active about it and talked to doctors or government sources, whereas others heard about COVID-19 from sources they engage with anyway, namely, social media and personal relationships. Those who were interested in COVID-19 due to fear or other reasons looked into the topic, gained a lot of knowledge, and were more likely to engage with the recommended behaviors regardless of their source. I think the authors should be more careful when stating their conclusions.

-> Thank you for this comment. We made revisions throughout the Discussion section (lines 245-291) to present the findings more conservatively. 

13. While underrepresented demographic groups are certainly a limitation, the issue of representativeness of social media users seems more relevant here. The sample included Facebook users who are likely interested in COVID-19, otherwise, they wouldn’t have engaged with the study (this explains the high knowledge scores). This is likely to affect the results (e.g., people who turn to religious figures are less likely to participate, and if they do they probably do not represent other people who consult with religious authorities).

-> Thank you for this comment. We tried to improve our discussion by elaborating more on the limitation of the sampling strategy, including all the points you raised (lines 252 – 273).

Response to Reviewer #2

1. This study shows the important role of COVID-19 information sources in affecting 30 people’s engagement in recommended protective behaviors. The idea is well described theorotically and analytically.

-> Thank you for this positive feedback!

Response to Reviewer #3

1. While the findings of the study are interesting and comprehensive, one major shortcoming of the statistical analysis is that the authors have used inferential methods on the basis of a non-probability sample (convenience sample). As inferences are often drawn / valid only if a probability sample is used, so the authors need to justify it. If authors have followed a previous work where inferences have been drawn on the basis of a non-probability sample, they must cite it. The authors must clear this point.

->Thank you for your comment. We improve the discussion by further emphasizing the limitation of the sampling strategy, specifically including the points you raised in your comment (lines 252 – 273). We also reviewed the entire manuscript to make sure that there is no inference made to the general US population based on our findings. We believe that our findings are relevant in a way that it suggests policymakers to re-consider their risk communication strategies, especially for those who use social media platforms as their primary source of information. We hope that our revisions sufficiently address your concerns. 

2. Moreover, the abstract has been divided into sub-sections. I suggest the authors to write a brief abstract as a single section. No need to divide the abstract into sub-sections.

->We followed your suggestion and removed the subsections in the abstract. 

3. Further, in the text, citations have been given using parenthesis. Use square brackets for citations.

->We updated the citations using EndNote’s PLoS output style to comply with the journal requirements.

Response to Reviewer #4

1. The paper addresses a "hot" topic in the context of the new coronavirus pandemic. The introduction provides even some information related to the literature review, even though the authors should try to better present the current state in a more critical manner. 

-> Thank you for this comment. We updated the introduction with additional paragraph (line 71 – 76) in order to further highlight the timeliness and significance of this paper. We hope these revisions help presenting the context in a more critical manner. 

2. The methodology is quite simple as the data used in the paper is taken by the authors from external sources. The authors provide information related to predictor variables and moderator variables, but no scheme of the proposed model is provided. In my opinion, with given information, it is hard for the work to be reproduced or adapted to similar situations. 

-> Thank you for this comment. In order to enhance the reproducibility, we added the equation for the regression analysis in the manuscript (lines 141-142) and included the survey questions used for the predictor and outcome variables in the Supporting Information file (Table S1). 

3. The results section contains little information strictly related to the subject. The authors provide a long table containing descriptive statistics related to the participants to the study, with no real connection to the results as the results are very brief with respect to this information. 

-> Thank you for this comment. We included Table 1 to comply with the standard of quantitative research papers suggested by APA (https://apastyle.apa.org/jars/quant-table-1.pdf). Following your comment, we excluded the variables that are not included in the final regression model from Table 1. Results pertinent to the effect modification is presented in Table 2, Figure 1, and the paragraph that is presented in lines 185-205. 

4. Figure 1 is hard to understand - can you please explain it more in the main body of the paper? 

-> Thank you for this comment. We provided an explanation of Figure 1 in the Results section, lines 200-205. 

5. The moderator variable used in the study is a variable from a list. In most of the cases, it might happen that a person can be connected to multiple sources of information, each of them bringing its own contribution to the result. I think that a more-in-depth analysis is needed.

-> Thank you for this comment. In this study we aimed to test the hypothesis that the “primary” source of information plays a significant moderating role in the association between knowledge and behavior. The questionnaire inquired the participants’ “primary” source of information. And, we controlled for the confounding role of the multiple sources of information used by respondents by adding a continuous variable on the “total number of sources used” as a covariate in the analysis. We agree that future studies are warranted to explore the complex relationship between multiple sources of information and its role in moderating the association between knowledge and behavior.

---

## [Decision Letter · Decision Letter 1]

25 Oct 2021

PONE-D-21-24232R1Impact of COVID-19-Related Knowledge on Protective Behaviors: The Moderating Role of Primary Sources of InformationPLOS ONE

Dear Dr. Tozan,

Thank you for submitting your manuscript to PLOS ONE. After careful consideration, we feel that it has merit but does not fully meet PLOS ONE’s publication criteria as it currently stands. Therefore, we invite you to submit a revised version of the manuscript that addresses the points raised during the review process.

The revised version of the paper has considered the reviewers' comments. There are still some aspects to be addressed as it results from the reviewers' comments. Please consider them when submitting the revised version.

We look forward to receiving your revised manuscript.

Kind regards,

Camelia Delcea

Academic Editor

PLOS ONE

Journal Requirements:

Reviewers' comments:

Reviewer's Responses to Questions

**Comments to the Author**

1. If the authors have adequately addressed your comments raised in a previous round of review and you feel that this manuscript is now acceptable for publication, you may indicate that here to bypass the “Comments to the Author” section, enter your conflict of interest statement in the “Confidential to Editor” section, and submit your "Accept" recommendation.

Reviewer #1: All comments have been addressed

Reviewer #3: All comments have been addressed

2. Is the manuscript technically sound, and do the data support the conclusions?

Reviewer #1: Yes

Reviewer #3: Yes

3. Has the statistical analysis been performed appropriately and rigorously? 

Reviewer #1: Yes

Reviewer #3: Yes

4. Have the authors made all data underlying the findings in their manuscript fully available?

Reviewer #1: Yes

Reviewer #3: Yes

5. Is the manuscript presented in an intelligible fashion and written in standard English?

Reviewer #1: Yes

Reviewer #3: Yes

6. Review Comments to the Author

Reviewer #1: The authors have successfully addressed most of my comments. I do have a few remaining questions.

1. The authors focus on primary information sources rather than number or type of sources. However, their introduction section does not mention primary information source (but rather, social media as a complementary source). I think the authors should explain why they chose to only focus on one source, what is the logical or theoretical justification for the decision.

2. Thank you for conducting the Rasch analysis. While your results support the use of a Rasch scale over summary scores given the great fit statistics, I understand it might be too difficult to change the analyses at this stage.

3. I understand the purpose of a Wilcoxon test, but the authors did not provide evidence that the data violate the assumptions of a t-test. They should add the sentence about the scores being left-skewed to the manusctipt.

4. The tables do not follow APA style.

Reviewer #3: COVID-19 is a global issue and these types of research studies are badly needed. I found this research study very interesting and worth publishable.

7. PLOS authors have the option to publish the peer review history of their article (what does this mean?). If published, this will include your full peer review and any attached files.

Reviewer #1: No

Reviewer #3: No

---

## [Author Response · Author response to Decision Letter 1]

27 Oct 2021

Reviewer #1: The authors have successfully addressed most of my comments. I do have a few remaining questions.

1. The authors focus on primary information sources rather than number or type of sources. However, their introduction section does not mention primary information source (but rather, social media as a complementary source). I think the authors should explain why they chose to only focus on one source, what is the logical or theoretical justification for the decision.

-> Thank you for this comment. We expanded the text in Introduction to explain further and justify the focus on primary source of information along with references from the published literature. Please kindly refer to lines 68-72 and lines 82-84. 

2. Thank you for conducting the Rasch analysis. While your results support the use of a Rasch scale over summary scores given the great fit statistics, I understand it might be too difficult to change the analyses at this stage.

-> Thank you for this feedback. 

3. I understand the purpose of a Wilcoxon test, but the authors did not provide evidence that the data violate the assumptions of a t-test. They should add the sentence about the scores being left-skewed to the manuscript.

-> Thank you for this feedback. We added this explanation in lines 136-137 as follows: “Pairwise Wilcoxon rank-sum test—a nonparametric alternative to the t test—was used, given the left-skewedness of the scores distribution, to compare the distribution of knowledge and behavior scores between each group.” 

4. The tables do not follow APA style.

-> Thank you for this feedback. We followed the SAMPL guideline (https://www.equator-network.org/wp-content/uploads/2013/03/SAMPL-Guidelines-3-13-13.pdf) for statistical reporting and PLOS ONE’s guideline (https://journals.plos.org/plosone/s/tables) for table formatting. We re-looked at both references to ensure correct reporting format and style and made following additional updates on both Table 1 and Table 2:

1) We added extra column, instead of using indents, to differentiate the variable name and subsequent category names.

2) We reported all the p-values equal to or greater than 0.001 as equalities and only reported values under 0.001 as inequalities.

3) Following your feedback, we formatted tables with the horizontal lines only to be compliant with APA style (https://apastyle.apa.org/style-grammar-guidelines/tables-figures/tables) 

4) To improve legibility, we used bold text for the variable names and primary column titles.

Reviewer #3: COVID-19 is a global issue and these types of research studies are badly needed. I found this research study very interesting and worth publishable.

->We thank you for your endorsement.

---

## [Decision Letter · Decision Letter 2]

15 Nov 2021

Impact of COVID-19-Related Knowledge on Protective Behaviors: The Moderating Role of Primary Sources of Information

PONE-D-21-24232R2

Dear Dr. Tozan,

We’re pleased to inform you that your manuscript has been judged scientifically suitable for publication and will be formally accepted for publication once it meets all outstanding technical requirements.

Kind regards,

Camelia Delcea

Academic Editor

PLOS ONE

Additional Editor Comments (optional):

Reviewers' comments:

Reviewer's Responses to Questions

**Comments to the Author**

1. If the authors have adequately addressed your comments raised in a previous round of review and you feel that this manuscript is now acceptable for publication, you may indicate that here to bypass the “Comments to the Author” section, enter your conflict of interest statement in the “Confidential to Editor” section, and submit your "Accept" recommendation.

Reviewer #1: All comments have been addressed

2. Is the manuscript technically sound, and do the data support the conclusions?

Reviewer #1: Yes

3. Has the statistical analysis been performed appropriately and rigorously? 

Reviewer #1: Yes

4. Have the authors made all data underlying the findings in their manuscript fully available?

Reviewer #1: Yes

5. Is the manuscript presented in an intelligible fashion and written in standard English?

Reviewer #1: Yes

6. Review Comments to the Author

Reviewer #1: The authors have addressed all of the comments. Good luck!

7. PLOS authors have the option to publish the peer review history of their article (what does this mean?). If published, this will include your full peer review and any attached files.

Reviewer #1: No

---

## [Editor Report · Acceptance letter]

17 Nov 2021

PONE-D-21-24232R2 

Impact of COVID-19-related knowledge on protective behaviors: The moderating role of primary sources of information 

Dear Dr. Tozan:

I'm pleased to inform you that your manuscript has been deemed suitable for publication in PLOS ONE. Congratulations! Your manuscript is now with our production department. 

Kind regards, 

on behalf of

Dr. Camelia Delcea 

Academic Editor

PLOS ONE